# Are Disentangled Representations Helpful for Abstract Visual Reasoning?

**Sjoerd van Steenkiste**
IDSIA, USI, SUPSI
sjoerd@idsia.ch

**Francesco Locatello**
ETH Zurich, MPI-IS
locatelf@ethz.ch

**Jürgen Schmidhuber**
IDSIA, USI, SUPSI, NNAISENSE
juergen@idsia.ch

**Olivier Bachem**
Google Research, Brain Team
bachem@google.com

## Abstract

A disentangled representation encodes information about the salient factors of variation in the data independently. Although it is often argued that this representational format is useful in learning to solve many real-world down-stream tasks, there is little empirical evidence that supports this claim. In this paper, we conduct a large-scale study that investigates whether disentangled representations are more suitable for abstract reasoning tasks. Using two new tasks similar to Raven's Progressive Matrices, we evaluate the usefulness of the representations learned by 360 state-of-the-art unsupervised disentanglement models. Based on these representations, we train 3600 abstract reasoning models and observe that disentangled representations do in fact lead to better down-stream performance. In particular, they enable quicker learning using fewer samples.

## 1 Introduction

Learning good representations of high-dimensional sensory data is of fundamental importance to Artificial Intelligence [4, 3, 6, 49, 7, 69, 67, 50, 59, 73]. In the supervised case, the quality of a representation is often expressed through the ability to solve the corresponding down-stream task. However, in order to leverage vasts amounts of unlabeled data, we require a set of desiderata that apply to more general real-world settings.

Following the successes in learning *distributed* representations that efficiently encode the content of high-dimensional sensory data [45, 56, 76], recent work has focused on learning representations that are *disentangled* [6, 69, 68, 73, 71, 26, 27, 42, 10, 63, 16, 52, 53, 48, 9, 51]. A disentangled representation captures information about the salient (or explanatory) factors of variation in the data, isolating information about each specific factor in only a few dimensions. Although the precise circumstances that give rise to disentanglement are still being debated, the core concept of a local correspondence between data-generative factors and learned latent codes is generally agreed upon [16, 26, 52, 63, 71].

Disentanglement is mostly about *how* information is encoded in the representation, and it is often argued that a representation that is disentangled is desirable in learning to solve challenging real-world down-stream tasks [6, 73, 59, 7, 26, 68]. Indeed, in a disentangled representation, information about an individual factor value can be readily accessed and is robust to changes in the input that do not affect this factor. Hence, learning to solve a down-stream task from a disentangled representation is expected to require fewer samples and be easier in general [68, 6, 28, 29, 59]. Real-world generative processes are also often based on latent spaces that factorize. In this case, a disentangled

representation that captures this product space is expected to help in generalizing systematically in this regard [18, 22, 59].

Several of these purported benefits can be traced back to empirical evidence presented in the recent literature. Disentangled representations have been found to be more sample-efficient [29], less sensitive to nuisance variables [55], and better in terms of (systematic) generalization [1, 16, 28, 35, 70]. However, in other cases it is less clear whether the observed benefits are actually due to disentanglement [48]. Indeed, while these results are generally encouraging, a systematic evaluation on a complex down-stream task of a wide variety of disentangled representations obtained by training different models, using different hyper-parameters and data sets, appears to be lacking.

**Contributions**  In this work, we conduct a large-scale evaluation[1] of disentangled representations to systematically evaluate some of these purported benefits. Rather than focusing on a simple single factor classification task, we evaluate the usefulness of disentangled representations on abstract visual reasoning tasks that challenge the current capabilities of state-of-the-art deep neural networks [30, 65]. Our key contributions include:

- We create two new visual abstract reasoning tasks similar to Raven's Progressive Matrices [61] based on two disentanglement data sets: *dSprites* [27], and *3dshapes* [42]. A key design property of these tasks is that they are hard to solve based on statistical co-occurrences and require reasoning about the relations between different objects.

- We train 360 unsupervised disentanglement models spanning four different disentanglement approaches on the individual images of these two data sets and extract their representations. We then train 3600 Wild Relation Networks [65] that use these disentangled representations to perform abstract reasoning and measure their accuracy at various stages of training.

- We evaluate the usefulness of disentangled representations by comparing the accuracy of these abstract reasoning models to the degree of disentanglement of the representations (measured using five different disentanglement metrics). We observe compelling evidence that more disentangled representations yield better sample-efficiency in learning to solve the considered abstract visual reasoning tasks. In this regard our results are complementary to a recent prior study of disentangled representations that did not find evidence of increased sample efficiency on a much simpler down-stream task [52].

## 2   Background and Related Work on Learning Disentangled Representations

Despite an increasing interest in learning disentangled representations, a precise definition is still a topic of debate [16, 26, 52, 63]. In recent work, Eastwood et al. [16] and Ridgeway et al. [63] put forth three criteria of disentangled representations: *modularity*, *compactness*, and *explicitness*. Modularity implies that each code in a learned representation is associated with only one factor of variation in the environment, while compactness ensures that information regarding a single factor is represented using only one or few codes. Combined, modularity and compactness suggest that a disentangled representation implements a one-to-one mapping between salient factors of variation in the environment and the learned codes. Finally, a disentangled representation is often assumed to be explicit, in that the mapping between factors and learned codes can be implemented with a simple (i.e. linear) model. While modularity is commonly agreed upon, compactness is a point of contention. Ridgeway et al. [63] argue that some features (eg. the rotation of an object) are best described with multiple codes although this is essentially not compact. The recent work by Higgins et al. [26] suggests an alternative view that may resolve these different perspectives in the future.

**Metrics**  Multiple metrics have been proposed that leverage the ground-truth generative factors of variation in the data to measure disentanglement in learned representations. In recent work, Locatello et al. [52] studied several of these metrics, which we will adopt for our purposes in this work: the *BetaVAE* score [27], the *FactorVAE* score [42], the *Mutual Information Gap (MIG)* [10], the disentanglement score from Eastwood et al. [16] referred to as the *DCI Disentanglement* score, and the *Separated Attribute Predictability (SAP)* score [48].

The *BetaVAE* score, *FactorVAE* score, and *DCI Disentanglement* score focus primarily on *modularity*. The former assess this property through *interventions*, i.e. by keeping one factor fixed and varying all others, while the *DCI Disentanglement* score estimates this property from the relative importance assigned to each feature by a random forest regressor in predicting the factor values. The *SAP* score and *MIG* are mostly focused on *compactness*. The *SAP* score reports the difference between the top two most predictive latent codes of a given factor, while *MIG* reports the difference between the top two latent variables with highest mutual information to a certain factor.

The degree of *explicitness* captured by any of the disentanglement metrics remain unclear. In prior work it was found that there is a positive correlation between disentanglement metrics and down-stream performance on single factor classification [52]. However, it is not obvious whether disentangled representations are useful for down-stream performance per se, or if the correlation is driven by the explicitness captured in the scores. In particular, the *DCI Disentanglement* score and the *SAP* score compute disentanglement by training a classifier on the representation. The former uses a random forest regressor to determine the relative importance of each feature, and the latter considers the gap in prediction accuracy of a support vector machine trained on each feature in the representation. *MIG* is based on the matrix of pairwise mutual information between factors of variations and dimensions of the representation, which also relates to the explicitness of the representation. On the other hand, the *BetaVAE* and *FactorVAE* scores predict the index of a fixed factor of variation and not the exact value.

We note that current disentanglement metrics each require access to the ground-truth factors of variation, which may hinder the practical feasibility of learning disentangled representations. Here our goal is to assess the usefulness of disentangled representations more generally (i.e. assuming it is possible to obtain them), which can be verified independently.

**Methods** Several methods have been proposed to learn disentangled representations. Here we are interested in evaluating the benefits of disentangled representations that have been learned through *unsupervised* learning. In order to control for potential confounding factors that may arise in using a single model, we use the representations learned from four state-of-the-art approaches from the literature: $\beta$-VAE [27], *FactorVAE* [42], $\beta$-TCVAE [10], and *DIP-VAE* [48]. A similar choice of models was used in a recent study by Locatello et al. [52].

Using notation from Tschannen et al. [73], we can view all of these models as Auto-Encoders that are trained with the regularized variational objective of the form:

$$\mathbb{E}_{p(x)}[\mathbb{E}_{q_\phi(z|x)}[-\log p_\theta(x|z)]] + \lambda_1 \mathbb{E}_{p(x)}[R_1(q_\phi(z|x))] + \lambda_2 R_2(q_\phi(z)). \tag{1}$$

The output of the *encoder* that parametrizes $q_\phi(z|x)$ yields the representation. Regularization serves to control the information flow through the bottleneck induced by the encoder, while different regularizers primarily vary in the notion of disentanglement that they induce. $\beta$-VAE restricts the capacity of the information bottleneck by penalizing the KL-divergence, using $\beta = \lambda_1 > 1$ with $R_1(q_\phi(z|x)) := D_{KL}[q_\phi(z|x)||p(z)]$, and $\lambda_2 = 0$; *FactorVAE* penalizes the Total Correlation [77] of the latent variables via adversarial training, using $\lambda_1 = 0$ and $\lambda_2 = 1$ with $R_2(q_\phi(z)) := TC(q_\phi(z))$; $\beta$-TCVAE also penalizes the Total Correlation but estimates its value via a biased Monte Carlo estimator; and finally *DIP-VAE* penalizes a mismatch in moments between the aggregated posterior and a factorized prior, using $\lambda_1 = 0$ and $\lambda_2 \geq 1$ with $R_2(q_\phi(z)) := ||\text{Cov}_{q_\phi(z)} - I||_F^2$.

**Other Related Works** Learning disentangled representations is similar in spirit to non-linear ICA, although it relies primarily on (architectural) inductive biases and different degrees of supervision [13, 2, 39, 36, 37, 38, 25, 33, 32]. Due to the initial poor performance of purely unsupervised methods, the field initially focused on semi-supervised [62, 11, 57, 58, 44, 46] and weakly supervised approaches [31, 12, 40, 21, 78, 20, 15, 35, 80, 54, 47, 64, 8]. In this paper, we consider the setup of the recent unsupervised methods [27, 26, 48, 42, 9, 52, 71, 10]. Finally, while this paper focuses on evaluating the benefits of disentangled *features*, these are complementary to recent work that focuses on the unsupervised "disentangling" of images into compositional primitives given by object-like representations [17, 23, 24, 22, 60, 74, 75]. Disentangling pose, style, or motion from content are classical vision tasks that has been studied with different degrees of supervision [72, 79, 80, 34, 19, 14, 21, 36].

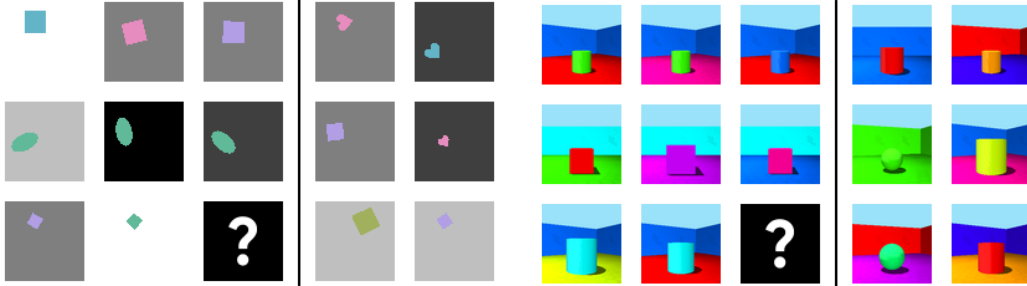

Figure 1: Examples of RPM-like abstract visual reasoning tasks using *dSprites* (left) and *3dshapes* (right). The correct answer and additional samples are available in Figure 17 in Appendix C.

## 3 Abstract Visual Reasoning Tasks for Disentangled Representations

In this work we evaluate the purported benefits of disentangled representations on abstract visual reasoning tasks. Abstract reasoning tasks require a learner to infer abstract relationships between multiple entities (i.e. objects in images) and re-apply this knowledge in newly encountered settings [41]. Humans are known to excel at this task, as is evident from experiments with simple visual IQ tests such as Raven's Progressive Matrices (RPMs) [61]. An RPM consists of several context panels organized in multiple sequences, with one sequence being incomplete. The task consists of completing the final sequence by choosing from a given set of answer panels. Choosing the correct answer panel requires one to infer the relationships between the panels in the complete context sequences, and apply this knowledge to the remaining partial sequence.

In recent work, Santoro et al. [65] evaluated the abstract reasoning capabilities of deep neural networks on this task. Using a data set of RPM-like matrices they found that standard deep neural network architectures struggle at abstract visual reasoning under different training and generalization regimes. Their results indicate that it is difficult to solve these tasks by relying purely on superficial image statistics, and can only be solved efficiently through abstract visual reasoning. This makes this setting particularly appealing for investigating the benefits of disentangled representations.

**Generating RPM-like Matrices**   Rather than evaluating disentangled representations on the Procedurally Generated Matrices (PGM) dataset from Barrett et al. [65] we construct two new abstract RPM-like visual reasoning datasets based on two existing datasets for disentangled representation learning. Our motivation for this is twofold: it is not clear what a ground-truth disentangled representation should look like for the PGM dataset, while the two existing disentanglement data sets include the ground-truth factors of variation. Secondly, in using established data sets for disentanglement, we can reuse hyper-parameter ranges that have proven successful. We note that our study is substantially different to recent work by Steenbrugge et al. [70] who evaluate the representation of a single trained $\beta$-VAE [27] on the original PGM data set.

To construct the abstract reasoning tasks, we use the ground-truth generative model of the *dSprites* [27] and *3dshapes* [42] data sets with the following changes[2]: For *dSprites*, we ignore the orientation feature for the abstract reasoning tasks as certain objects such as squares and ellipses exhibit rotational symmetries. To compensate, we add background color (5 different shades of gray linearly spaced between white and black) and object color (6 different colors linearly spaced in HUSL hue space) as two new factors of variation. Similarly, for the abstract reasoning tasks (but not when learning representations), we only consider three different values for the scale of the object (instead of 6) and only four values for the x and y position (instead of 32). For *3dshapes*, we retain all of the original factors but only consider four different values for scale and azimuth (out of 8 and 16) for the abstract reasoning tasks. We refer to Figure 7 in Appendix B for samples from these data sets.

For the modified *dSprites* and *3dshapes*, we now create corresponding abstract reasoning tasks. The key idea is that one is given a $3 \times 3$ matrix of context image panels with the bottom right image panel missing, as well as a set of six potential answer panels (see Figure 1 for an example). One then has to infer which of the answers fits in the missing panel of the $3 \times 3$ matrix based on relations between

image panels in the rows of the $3 \times 3$ matrices. Due to the categorical nature of ground-truth factors in the underlying data sets, we focus on the *AND* relationship in which one or more factor values are equal across a sequence of context panels [65].

We generate instances of the abstract reasoning tasks in the following way: First, we uniformly sample whether 1, 2, or 3 ground-truth factors are fixed across rows in the instance to be generated. Second, we uniformly sample without replacement the set of underlying factors in the underlying generative model that should be kept constant. Third, we uniformly sample a factor value from the ground-truth model for each of the three rows and for each of the fixed factors[3]. Fourth, for all other ground-truth factors we also sample $3 \times 3$ matrices of factor values from the ground-truth model with the single constraint that the factor values are not allowed to be constant across the first two rows (in that case we sample a new set of values). After this we have ground-truth factor values for each of the 9 panels in the correct solution to the abstract reasoning task, and we can sample corresponding images from the ground-truth model. To generate difficult alternative answers, we take the factor values of the correct answer panel and randomly resample the non-fixed factors as well as a random fixed factor until the factor values no longer satisfy the relations in the original abstract reasoning task. We repeat this process to obtain five incorrect answers and finally insert the correct answer in a random position. Examples of the resulting abstract reasoning tasks can be seen in Figure 1 as well as in Figures 18 and 19 in Appendix C.

**Models**   We will make use of the *Wild Relation Network (WReN)* to solve the abstract visual reasoning tasks [65]. It incorporates relational structure, and was introduced in prior work specifically for such tasks. The *WReN* is evaluated for each answer panel $a \in A = \{a_1, ..., a_6\}$ in relation to all the context-panels $C = \{c_1, ..., c_8\}$ as follows:

$$\text{WReN}(a, C) = f_\phi( \sum_{e_1, e_2 \in E} g_\theta(e_1, e_2)) \ , \ E = \{\text{CNN}(c_1), ..., \text{CNN}(c_8)\} \cup \{\text{CNN}(a)\} \quad (2)$$

First an embedding is computed for each panel using a deep Convolutional Neural Network (CNN), which serve as input to a Relation Network (RN) module [66]. The Relation Network reasons about the different relationships between the context and answer panels, and outputs a score. The answer panel $a \in A$ with the highest score is chosen as the final output.

The Relation Network implements a suitable inductive bias for (relational) reasoning [5]. It separates the reasoning process into two stages. First $g_\theta$ is applied to all pairs of panel embeddings to consider relations between the answer panel and each of the context panels, and relations among the context panels. Weight-sharing of $g_\theta$ between the panel-embedding pairs makes it difficult to overfit to the image statistics of the individual panels. Finally, $f_\phi$ produces a score for the given answer panel in relation to the context panels by globally considering the different relations between the panels as a whole. Note that in using the same *WReN* for different answer panels it is ensured that each answer panel is subject to the same reasoning process.

## 4   Experiments

### 4.1   Learning Disentangled Representations

We train $\beta$-*VAE* [27], *FactorVAE* [42], $\beta$-*TCVAE* [10], and *DIP-VAE* [48] on the panels from the modified *dSprites* and *3dshapes* data sets[4]. For $\beta$-*VAE* we consider two variations: the standard version using a fixed $\beta$, and a version trained with the *controlled capacity increase* presented by Burgess et al. [9]. Similarly for *DIP-VAE* we consider both the *DIP-VAE-I* and *DIP-VAE-II* variations of the proposed regularizer [48]. For each of these methods, we considered six different values for their (main) hyper-parameter and five different random seeds. The remaining experimental details are presented in Appendix A.

After training, we end up with 360 encoders, whose outputs are expected to cover a wide variation of different representational formats with which to encode information in the images. Figures 9 and 10 in the Appendix show histograms of the reconstruction errors obtained after training, and

the scores that various disentanglement metrics assigned to the corresponding representations. The reconstructions are mostly good (see also Figure 7), which confirms that the learned representations tend to accurately capture the image content. Correspondingly, we expect any observed difference in down-stream performance when using these representations to be primarily the result of *how* information is encoded. In terms of the scores of the various disentanglement metrics, we observe a wide range of values. It suggests that in going by different definitions of disentanglement, there are large differences among the quality of the learned representations.

## 4.2 Abstract Visual Reasoning

We train different WReN models where we control for two potential confounding factors: the representation produced by a specific model used to embed the input images, as well as the hyper-parameters of the WReN model. For hyper-parameters, we use a random search space as specified in Appendix A. We used the following training protocol: We train each of these models using a batch size of 32 for 100K iterations where each mini-batch consists of *newly generated* random instances of the abstract reasoning tasks. Similarly, every 1000 iterations, we evaluate the accuracy on 100 mini-batches of fresh samples. We note that this corresponds to the statistical optimization setting, sidestepping the need to investigate the impact of empirical risk minimization and overfitting[5].

### 4.2.1 Initial Study

First, we trained a set of baseline models to assess the overall complexity of the abstract reasoning task. We consider three types of representations: (i) CNN representations which are learned from scratch (with the same architecture as in the disentanglement models) yielding standard WReN, (ii) pre-trained frozen representations based on a random selection of the pre-trained disentanglement models, and (iii) directly using the ground-truth factors of variation (both one-hot encoded and integer encoded). We train 30 different models for each of these approaches and data sets with different random seeds and different draws from the search space over hyper-parameter values.

An overview of the training behaviour and the accuracies achieved can be seen in Figures 2 and 11 (Appendix B). We observe that the standard WReN model struggles to obtain good results on average, even after having seen many different samples at 100K steps. This is due to the fact that training from scratch is hard and runs may get stuck in local minima where they predict each of the answers with equal probabilities. Given the pre-training and the exposure to additional unsupervised samples, it is not surprising that the learned representations from the disentanglement models perform better. The WReN models that are given the true factors also perform well, already after only few steps of training. We also observe that different runs exhibit a significant spread, which motivates why we analyze the average accuracy across many runs in the next section.

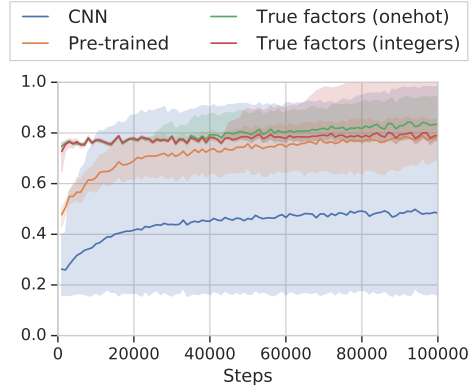

Figure 2: Average down-stream accuracy of baselines, and models using pre-trained representations on *dSprites*. Shaded area indicates min and max accuracy.

It appears that *dSprites* is the harder task, with models reaching an average score of $80\%$, while reaching an average of $90\%$ on *3dshapes*. Finally, we note that most learning progress takes place in the first 20K steps, and thus expect the benefits of disentangled representations to be most clear in this regime.

### 4.2.2 Evaluating Disentangled Representations

Based on the results from the initial study, we train a full set of WReN models in the following manner: We first sample a set of 10 hyper-parameter configurations from our search space and then trained WReN models using these configurations for each of the 360 representations from the disentanglement

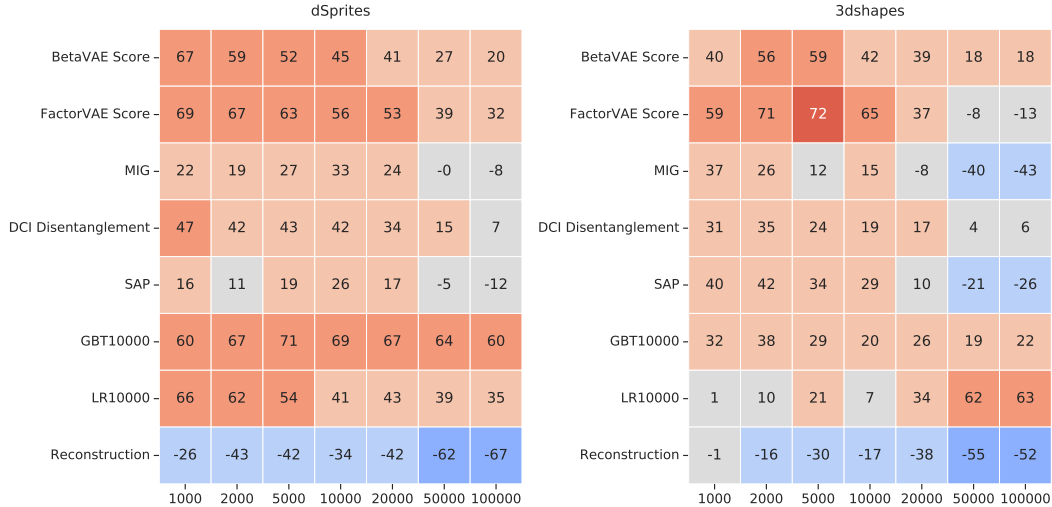

Figure 3: Rank correlation between various metrics and down-stream accuracy of the abstract visual reasoning models throughout training (i.e. for different number of samples).

models. We then compare the average down-stream training accuracy of WReN with the *BetaVAE* score, the *FactorVAE* score, *MIG*, the *DCI Disentanglement* score, and the *Reconstruction* error obtained by the decoder on the unsupervised learning task. As a sanity check, we also compare with the accuracy of a *Gradient Boosted Tree (GBT10000)* ensemble and a *Logistic Regressor (LR10000)* on single factor classification (averaged across factors) as measured on 10K samples. As expected, we observe a positive correlation between the performance of the WReN and the classifiers (see Figure 3).

**Differences in Disentanglement Metrics** Figure 3 displays the rank correlation (Spearman) between these metrics and the down-stream classification accuracy, evaluated after training for 1K, 2K, 5K, 10K, 20K, 50K, and 100K steps. If we focus on the disentanglement metrics, several interesting observations can be made. In the few-sample regime (up to 20K steps) and across both data sets it can be seen that both the *BetaVAE* score, and the *FactorVAE* score are highly correlated with down-stream accuracy. The *DCI Disentanglement* score is correlated slightly less, while the *MIG* and *SAP* score exhibit a relatively weak correlation.

These differences between the different disentanglement metrics are perhaps not surprising, as they are also reflected in their overall correlation (see Figure 8 in Appendix B). Note that the *BetaVAE* score, and the *FactorVAE* score directly measure the effect of intervention, i.e. what happens to the representation if all factors but one are varied, which is expected to be beneficial in efficiently comparing the content of two representations as required for this task. Similarly, it may be that *MIG* and *SAP* score have a more difficult time in differentiating representations that are only partially disentangled. Finally, we note that the best performing metrics on this task are mostly measuring *modularity*, as opposed to *compactness*. A more detailed overview of the correlation between the various metrics and down-stream accuracy can be seen in Figures 12 and 13 in Appendix B.

**Disentangled Representations in the Few-Sample Regime** If we compare the correlation of the disentanglement metric with the highest correlation (*FactorVAE*) to that of the *Reconstruction* error in the few-sample regime, then we find that disentanglement correlates much better with down-stream accuracy. Indeed, while low *Reconstruction* error indicates that all information is available in the representation (to reconstruct the image) it makes no assumptions about *how* this information is encoded. We observe strong evidence that disentangled representations yield better down-stream accuracy using relatively few samples, and we therefore conclude that they are indeed more sample efficient compared to entangled representations in this regard.

Figure 4 demonstrates the down-stream accuracy of the WReNs throughout training, binned into quartiles according to their degree of being disentangled as measured by the *FactorVAE* score (left), and in terms of *Reconstruction* error (right). It can be seen that representations that are more disentangled give rise to better relative performance consistently throughout all phases of training. If

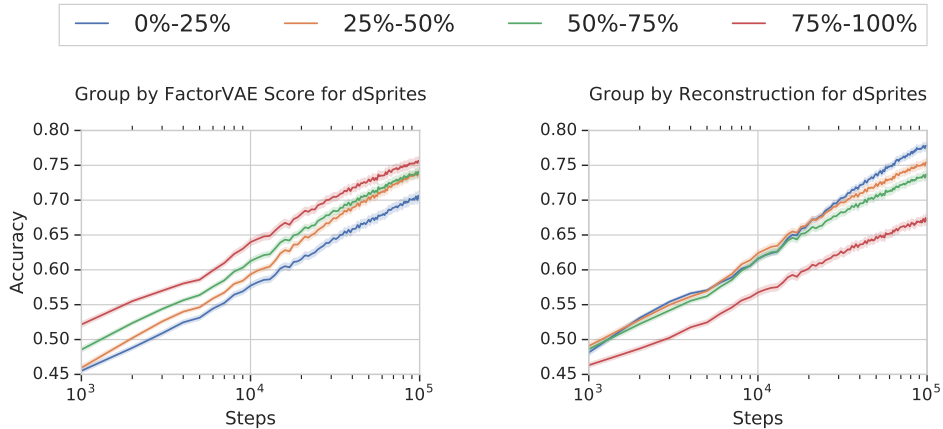

Figure 4: Down-stream accuracy of the WReN models throughout training, binned in quartiles based on the values assigned by the *FactorVAE* score (left), and *Reconstruction* error (right).

we group models according to their *Reconstruction* error then we find that this (reversed) ordering is much less pronounced. An overview for all other metrics can be seen in Figures 14 and 15.

**Disentangled Representations in the Many-Sample Regime**   In the many-sample regime (i.e. when training for 100K steps on batches of randomly drawn instances in Figure 3) we find that there is no longer a strong correlation between the scores assigned by the various disentanglement metrics and down-stream performance. This is perhaps not surprising as neural networks are general function approximators that, given access to enough labeled samples, are expected to overcome potential difficulties in using entangled representations. The observation that *Reconstruction* error correlates much more strongly with down-stream accuracy in this regime further confirms that this is the case.

A similar observation can be made if we look at the difference in down-stream accuracy between the top and bottom half of the models according to each metric in Figures 5 and 16 (Appendix B). For all disentanglement metrics, larger positive differences are observed in the few-sample regime that gradually reduce as more samples are observed. Meanwhile, the gap gradually increases for *Reconstruction* error upon seeing additional samples.

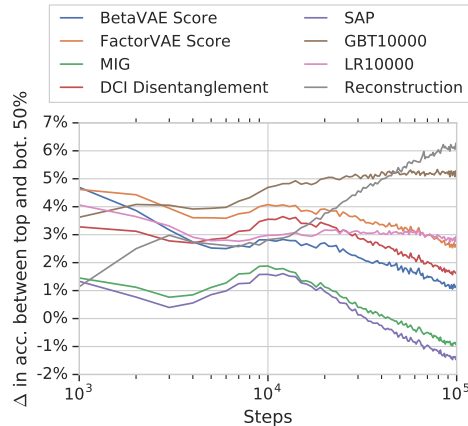

Figure 5: Difference in down-stream accuracy between top $50\%$ and bottom $50\%$, according to various metrics on *dSprites*.

**Differences in terms of Final Accuracy**   In our final analysis we consider the rank correlation between down-stream accuracy and the various metrics, split according to their final accuracy. Figure 6 shows the rank correlation for the worst performing fifty percent of the models after 100K steps (top), and for the best performing fifty percent (bottom). While these results should be interpreted with care as the split depends on the final accuracy, we still observe interesting results: It can be seen that disentanglement (i.e. *FactorVAE* score) remains strongly correlated with down-stream performance for both splits in the few-sample regime. At the same time, the benefit of lower *Reconstruction* error appears to be limited to the worst 50% of models. This is intuitive, as when the *Reconstruction* error is too high there may not be enough information present to solve the down-stream tasks. However, regarding the top performing models (best 50%), it appears that the *relative* gains from further reducing reconstruction error are of limited use.

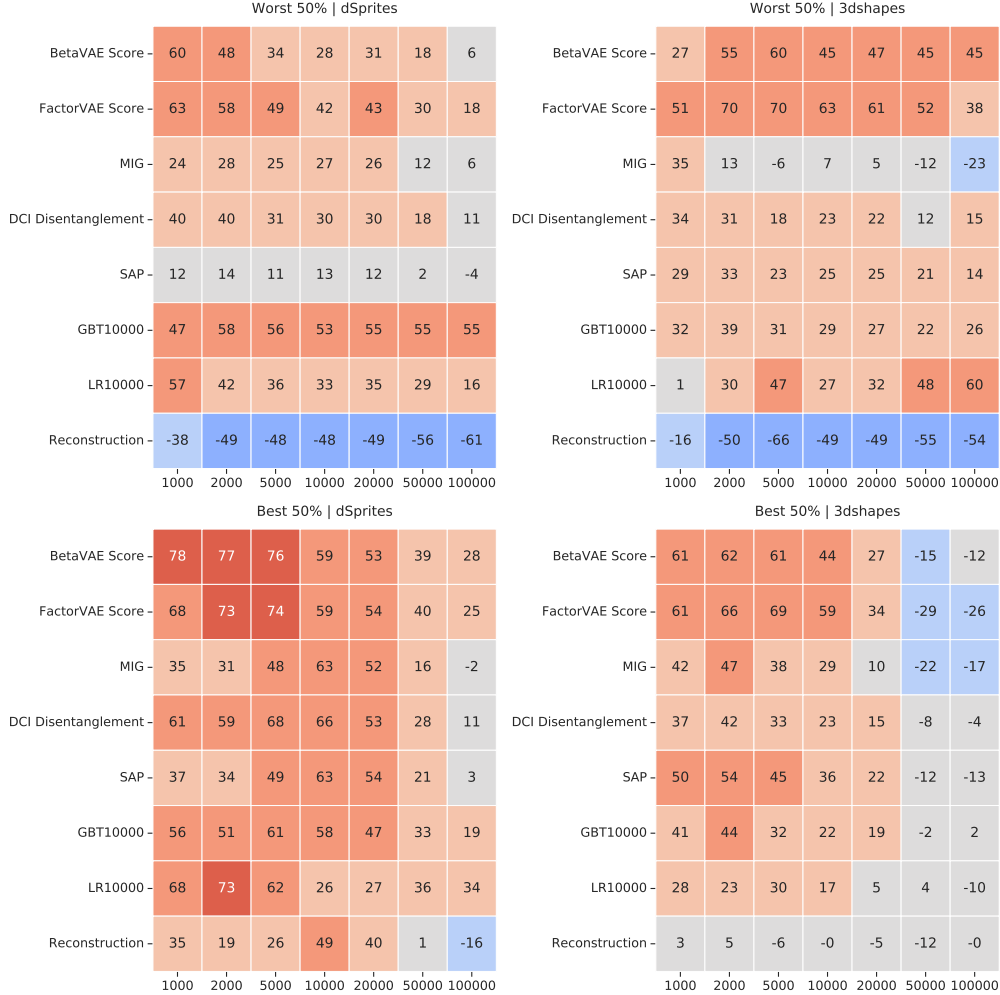

Figure 6: Rank correlation between various metrics and down-stream accuracy of the abstract visual reasoning models throughout training (i.e. for different number of samples). The results in the top row are based on the worst $50\%$ of the models (according to final accuracy), and those in the bottom row based on the best $50\%$ of the models. Columns correspond to different data sets.

# 5 Conclusion

In this work we investigated whether disentangled representations allow one to learn good models for non-trivial down-stream tasks with fewer samples. We created two abstract visual reasoning tasks based on existing data sets for which the ground truth factors of variation are known. We trained a diverse set of 360 disentanglement models based on four state-of-the-art disentanglement approaches and evaluated their representations using 3600 abstract reasoning models. We observed compelling evidence that more disentangled representations are more sample-efficient in the considered down-stream learning task. We draw three main conclusions from these results: First, these results provide concrete motivation why one might want to pursue disentanglement as a property of learned representations in the unsupervised case. Second, we still observed differences between disentanglement metrics, which should motivate further work in understanding what different properties they capture. None of the metrics achieved perfect correlation in the few-sample regime, which also suggests that it is not yet fully understood what makes one representation better than another in terms of learning. Third, it might be useful to extend the methodology in this study to other complex down-stream tasks, or include an investigation of other purported benefits of disentangled representations.

## Acknowledgments

The authors thank Adam Santoro, Josip Djolonga, Paulo Rauber and the anonymous reviewers for helpful discussions and comments. This research was partially supported by the Max Planck ETH Center for Learning Systems, a Google Ph.D. Fellowship (to Francesco Locatello), and the Swiss National Science Foundation (grant 200021_165675/1 to Jürgen Schmidhuber). This work was partially done while Francesco Locatello was at Google Research.

## Footnotes

[1]Reproducing these experiments requires approximately 2.73 GPU years (NVIDIA P100).

[2]These were implemented to ensure that humans can visually distinguish between the different values of each factor of variation.

[3]Note that different rows may have different values.

[4]Code is made available as part of *disentanglement_lib* at `https://git.io/JelEv`.

[5]Note that the state space of the data generating distribution is very large: $10^6$ factor combinations per panel and 14 panels for each instance yield more than $10^{144}$ potential instances (minus invalid configurations).

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
