[Supplementary Material · Abstract_Reasoning_via_Disentanglement - Supplementary.pdf]

# A Architectures and Hyper-parameters

## A.1 Disentanglement Methods

We use the same architecture, hyper-parameters and training setup as in prior work [52], which we report here for completeness. The architecture is depicted in Table 1. All models share the following hyper-parameters: We used a batch size of 64, 10-dimensional latent space and Bernoulli decoders. We trained the models for 300K steps using the Adam optimizer [43] with $\beta_1 = 0.9$, $\beta_2 = 0.999$, $\epsilon = 10^{-8}$ and a learning rate of 0.0001.

For $\beta$-*VAE*, we perform a sweep on $\beta$ on the interval $[1, 2, 4, 6, 8, 16]$. For $\beta$-*VAE* with *controlled capacity increase*, we perform a sweep on $c_{max}$ on the interval $[5, 10, 25, 50, 75, 100]$. The iteration threshold is set to 100K and $\gamma = 1000$. For *FactorVAE*, we perform a sweep on $\gamma$ on the interval $[10, 20, 30, 40, 50, 100]$. For the discriminator of the *FactorVAE* we use the architecture described in Table 2. Its other hyper-parameters are: Batch size = 64, Optimizer = Adam with $\beta_1 = 0.5$, $\beta_2 = 0.9$, $\epsilon = 10^{-8}$, and learning rate = 0.0001. For *DIP-VAE-I*, we perform a sweep on $\lambda_{od}$ on the interval $[1, 2, 5, 10, 20, 50]$, and set $\lambda_d = 10$. For *DIP-VAE-II*, we perform a sweep on $\beta$ on the interval $[1, 2, 5, 10, 20, 50]$, and set $\lambda_d = 10$. For $\beta$-*TCVAE*, we perform a sweep on $\beta$ on the interval $[1, 2, 4, 6, 8, 10]$. Each model is trained using 5 different random seeds.

Table 1: Encoder and Decoder architectures.

| Encoder | Decoder |
|---|---|
| Input: $64 \times 64 \times$ number of channels | Input: $\mathbb{R}^{10}$ |
| $4 \times 4$ conv, 32 ReLU, stride 2 | FC, 256 ReLU |
| $4 \times 4$ conv, 32 ReLU, stride 2 | FC, $4 \times 4 \times 64$ ReLU |
| $4 \times 4$ conv, 64 ReLU, stride 2 | $4 \times 4$ upconv, 64 ReLU, stride 2 |
| $4 \times 4$ conv, 64 ReLU, stride 2 | $4 \times 4$ upconv, 32 ReLU, stride 2 |
| FC 256, FC $2 \times 10$ | $4 \times 4$ upconv, 32 ReLU, stride 2 |
| | $4 \times 4$ upconv, number of channels, stride 2 |

Table 2: Architecture for the discriminator in *FactorVAE*.

| Discriminator |
|---|
| FC, 1000 leaky ReLU |
| FC, 1000 leaky ReLU |
| FC, 1000 leaky ReLU |
| FC, 1000 leaky ReLU |
| FC, 1000 leaky ReLU |
| FC, 1000 leaky ReLU |
| FC, 2 |

## A.2 Abstract Visual Reasoning Method

To solve the abstract reasoning tasks, we implemented the *Wild Relation Networks (WReN)* model of Barrett et al. [65]. For the experiments, we use the following random search space over the hyper-parameters: We uniformly sample a learning rate for the Adam optimizer from the set $\{0.01, 0.001, 0.0001\}$ while $\beta_1 = 0.9$, $\beta_2 = 0.999$, and $\epsilon = 10^{-8}$. For the edge MLP $g$ in the *WReN* model, we uniformly choose either 256 or 512 hidden units and we uniformly sample whether it has 2, 3, or 4 hidden layers. Similarly, for the graph MLP $f$ in the *WReN* model, we uniformly choose either 128 or 256 hidden units and we uniformly sample whether it has 1 or 2 hidden layers before the final linear layer to compute the final score. We also uniformly sample whether we apply no dropout, dropout of 0.25, dropout of 0.5, or dropout of 0.75 to units before this last layer.

# B  Additional Results

## B.1  Additional Results of Representation Learning

This subsection contains additional results in evaluating the training of the 360 disentanglement models. Figure 7 presents example reconstructions for different data sets and models that are representative of the median reconstruction error. Figure 8 displays the rank correlation between the various metrics on the learned representations.

Finally, Figures 9 and 10 present histograms of the scores assigned by various metrics to the learned representations on *dSprites* and *3dshapes* respectively.

(a) DIP-VAE-I trained on *3dshapes*.　　　　　(b) FactorVAE trained on *dSprites*.

Figure 7: Reconstructions for different data sets and models (representative samples of median reconstruction error). Odd columns show real samples and even columns their reconstruction. *3dshapes* appears to be much easier than *dSprites* where disentangling the shape appears hard.

Figure 8: Rank correlations between the different metrics considered in this paper.

Figure 9: Distribution of scores assigned by various metrics to the learned representations on *dSprites*.

Figure 10: Distribution of scores assigned by various metrics to the learned representations on *3dshapes*.

## B.2  Additional Results of Abstract Visual Reasoning

This subsection contains additional results obtained after training 3600 WReN models on the down-stream abstract visual reasoning tasks. Figure 11 presents the results for the various baselines on *3dshapes*. Figures 12 and 13 provide an in-depth view of the correlation between the scores assigned by various metrics and the down-stream accuracy.

Figures 14 and 15 present the down-stream accuracy at various stages of training of models grouped in quartiles according to the scores assigned by a given metric on *dSprites* and *3dshapes* respectively. Figure 16 presents the difference in down-stream accuracy of the best $50\%$ and worst $50\%$ as determined by each metric throughout training on *3dshapes*.

Figure 11: Down-stream accuracy of baselines, and models using pre-trained representations on *3dshapes*. Shaded area indicates min / max.

Figure 12: Correlation between *BetaVAE* score, *FactorVAE* score, *MIG*, *DCI Disentanglement* score, and *SAP* score (rows) and down-stream accuracy of the abstract visual reasoning models. Columns correspond to 1K, 5K, 10K, 100K training steps (i.e. number of samples).

Figure 13: Correlation between *GBT10000*, *LR10000*, and *Reconstruction* error (rows) and downstream accuracy of the abstract visual reasoning models. Columns correspond to 1K, 5K, 10K, 100K training steps (i.e. number of samples).

Figure 14: Down-stream accuracy of abstract visual reasoning models on *dSprites* throughout training (i.e. for different number of samples) binned in quartiles based on different metrics.

Figure 15: Down-stream accuracy of abstract visual reasoning models on *3dshapes* throughout training (i.e. for different number of samples) binned in quartiles based on different metrics.

Figure 16: Difference in down-stream accuracy between top 50% and bottom 50%, according to various metrics on *3dshapes*. X-axis is in log scale.

# C   Abstract Visual Reasoning Data

Figure 17 contains the answers to the PGM-like abstract visual reasoning tasks on *dSprites* and *3dshapes*. Focusing on the right example in Figure 17, note that the correct answer cannot be found by only considering the incomplete sequence of the context panel and the answer panels. In particular, we can not tell whether 1, 2 or 3 relationships hold and if for example the wall color or the object color is constant. As a result, one must consider the other two rows of context panels to deduce that it is background color, the azimuth and the shape-type that are equal among the panels. Then, this insight needs to be applied to the bottom row to see that a cylinder, a specific view point, and a lighter blue background are required in the correct solution. Then, the single answer panel fulfilling these criteria need to be selected.

Figure 17: Answers to the examples of the RPM-like abstract visual reasoning tasks.

Figures 18 and 19 contain additional examples (including answers) of the visual reasoning tasks for each data set respectively.

Figure 18: Additional examples (including answers) of the RPM-like abstract visual reasoning task using *dSprites*.

Figure 19: Additional examples (including answers) of the RPM-like abstract visual reasoning task using *3dshapes*.