[Reviews · NeurIPS 2019]

Reviewer 1



Thanks to the authors for the response. I thought the paper would be really cool and instructive IF the abstract tasks made sense. Based on the author's response and R3's review, these are pretty standard tasks I guess. I don't know much about these RPM tasks, so I will downgrade my confidence to a 2. The example tasks are still a bit confusing to me (compared to if you just look at an RPM example on wikipedia), but I guess once you see the answer to a few, you get the gist of them. Moreover, intuitively it seems that you do need to represent the factors of variation to be good at these tasks. My other concern is the one R2 raised, which is that if you need ground truth to pick out a good disentangling model and if ground truth helps a lot for directly solving the disentangling tasks then 1) why do we need disentangling for AVR? and 2) how would a AVR practitioner without ground truth label information benefit from these results? I think that despite the issues with this approach, I agree with the authors' point that for the sake of the study we can suspend some skepticism about how we get to these representations. Also the most important point the authors raised was that it is of high relevance to validate the motivation of the 20+ preceding disentangling papers by actually measuring how well disentangled representations do on downstream tasks. I wholeheartedly agree (and think more people should be doing this), so I will upgrade to a 7 despite the flaws in the experimental setup. ----------------------------------------------------------- They formulate two abstract visual reasoning tasks based on dSprites and 3dshapes. They then see how well different disentangled representations do when transferred to these tasks by training a relational model on top of these representations. They find that disentangled representations result in more sample efficient transfer to these abstract visual reasoning tasks, whereas in high sample regimes disentangling does not correlate much with upstream accuracy. Strengths: * Well-written. Background and related work is really well explained * Really helpful to show reconstruction error’s correlation to the upstream tasks as it is often used as a proxy for good performance * A cool, instructive result that disentangling is sample efficient * An unsurprising, but also instructive result that any entangled representation that captures most of the data (low reconstruction error) does well in upstream task when one has a lot of labelled data. Weaknesses: * Really only one real takeaway/useful experiment from the paper, which is that disentangling is sample efficient for this strange set of upstream tasks. * I have a lot of problems with these abstract visual reasoning tasks. They seem a bit unintuitive and overly difficult (I have a lot of trouble solving them). Having multiple rows and having multiple and different factors changing between each frame is very confusing and it seems like it would be hard to interpret how much these models actually learn the pattern or just exploit some artifacts. Do we have any proof that more simpler visual reasoning tasks wouldn’t do and this formulation in the paper is the way to go? * It seems weird the authors didn’t just consider a task with one row and one panel missing and the same one factor changing between panels. Is there any empirical evidence that this is too easy or uninformative? Why not a row where there are a few panels of the ellipse getting bigger and then for the missing frame the model chooses between a smaller ellipse, same size ellipse, *bigger ellipse*, bigger ellipse but at the wrong angle, bigger ellipse, but translated, bigger ellipse but different color, etc. or at least some progression of difficulty starting from the easiest and working up to the tasks in the paper?

Reviewer 2



Originality This paper does not focus on developing a novel method. All disentanglement methods have been previously proposed. The WReN that solves the abstract reasoning tasks is also an existing method. Simply combining these methods does not seem novel. Quality I have concerns about the methodology adopted in this paper. The paper focuses on discussing the relationship between the accuracy of the abstract reasoning tasks and the disentanglement score. However, disentanglement scores can only be computed when the ground-truth factors of variation are available. If ground-truth factors are available, then we can directly use the ground-truth factors to train WReN and achieve excellent performance, as shown in Figure 2, or we can train regressors/classifiers that predict the ground-truth factor before training WReN; but we do not need disentanglement learning. If ground-truth factors are not available, then we can not compute disentanglement scores, and we are not able to utilize the results are shown in Figure 3, 4 and 5 to select the best disentangled representation. Therefore, It looks to me that disentanglement learning is not very helpful in abstract reasoning tasks. Clarity This paper is well-organized and not difficult to follow. Significance The details are provided in Section 1. I think the contribution of this paper would be reasonable, if the authors can address my concerns about the methodology. Minor issues It looks to me that the word "up-stream" in this paper should be changed to "down-stream"

Reviewer 3



The paper conducts a large-scale study of the performance of disentangled representations on upstream abstract reasoning tasks. The abstract reasoning tasks use the methodology of Raven’s progressive matrices but use samples from dSprites and 3dshapes as the skin, with some modifications. Wild Relation Network is used as the upstream model, which would use representations learned by the models under comparison: beta-vAE, FactorVAE, beta-TCVAE, DIP-VAE, and variants of these which improve on it. There are many small bits of useful information in the paper, such as the fact that metrics which measure modularity as opposed to compactness perform better in the upstream task. However, the main conclusion of the paper is that disentangled representations, in general, do enable sample efficient learning in low-sample regimes as compared to learning from scratch. I wish the analysis could have been clearer and mode space was dedicated to it. I don’t fully understand how gradient-boosted trees or logistic regression were used as points of comparison. The first three pages are not very information-dense and perhaps should be compressed so that we get to the good stuff faster. Similarly, small details about the dataset generation could have been moved to the appendix. However, overall the paper is well-written and my criticism on clarity is minor. The paper tackles a very important question on representation learning and provides interesting new insights about it.

[Author Response · NeurIPS 2019]

We thank the reviewers for their careful consideration of our paper and for the useful feedback. We are happy to see that the reviewers find that the paper is "*well-written*", contributes a "*cool, instructive result*", and that it "*tackles a very important question on representation learning and provides interesting new insights about it*". However, there appear two separate major concerns by R1 and R2 to which we kindly respond below.

**About the Abstract Visual Reasoning Task (R1)**

R1 raised the concern that the abstract visual reasoning tasks considered in this paper "... *seem a bit unintuitive and overly difficult*", finds it "... *weird the authors didn't just consider a task with one row and one panel missing and the same one factor changing between panels*", and requires "... *a very good explanation as to why the strange formulation in the paper was used and simpler tasks weren't used*". We will provide this explanation next.

- **Abstract visual reasoning to evaluate disentangled representations.** Disentanglement prescribes *how* information about salient factors of variation in the input should be encoded. Hence, we require a task that (1) involves reasoning about these factors, (2) that is well established and connects to the literature, and (3) that can not be trivially solved through correlating image statistics. This leads us to Raven's Progressive Matrices (RPMs; [59]).

- **RPMs are a standard test for human abstract reasoning.** $3 \times 3$ RPMs (standard setting - as in our paper) require one to identify relationships among salient factors of variation in two complete rows of images, and apply these relationships to answer a multiple choice question to complete the final panel of the third row. RPMs are a standard test to estimate abstract reasoning capabilities (Motta & Joseph, *Handbook of Psychological Assessment*, 2016).

- **RPMs are used in prior work in ML.** RPMs have been considered in prior work in ML, e.g. [30, 63, 68], and in the cognitive science literature, see (Lovett & Forbus, *Modeling visual problem solving as analogical reasoning*, 2017) for an overview. We created RPMs on two disentanglement data sets as closely as possible to the prior work.

- **RPMs cannot be solved trivially.** In recent work [63] the abstract reasoning capacity of several deep neural networks were tested on a data set consisting of $3 \times 3$ RPM-like abstract visual reasoning tasks. It was confirmed that this is a difficult task at which traditional architectures struggle, while the WReN architecture (that incorporates elements to enhance relational reasoning capabilities) performs best.

- **Validated through initial study & humans.** Our initial study in Section 4.2.1 validates that our adaptation of the RPM task serves as a sensible benchmark for disentangled representations, and we have further informally tested the task with >10 people. Finally, we note that we only consider the AND relationship, which is a simpler setting, and that the data sets considered in fact test two difficulties due to differences in their number of possible feature combinations.

**About Methodological Concerns (R2)**

R2 raised concerns about the methodology in this paper, arguing that "*If ground-truth factors are available, then we can directly use the ground-truth factors to train WReN and achieve excellent performance ... but we do not need disentanglement learning*, and that "*if ground-truth factors are not available, then we can not compute disentanglement scores, and we are not able to utilize the results are shown in Figure 3, 4 and 5 to select the best disentangled representation*". It appears that there is some confusion with regards to the goals and contributions of this paper.

- **Relevance: validate the motivation of >20 recent ML papers.** Recently, numerous papers have been concerned with learning disentangled representations [1, 8, 9, 10, 11, 15, 16, 17, 18, 20, 26, 27, 28, 29, 35, 42, 47, 51, 55, 56, 60, 61, 62, 76, 77]. The key motivation (but also assumption) of these works is that current notions of disentanglement (MIG, DCI, *etc.*) are desirable but until now there has been little empirical evidence verifying this.

- **How? By evaluating disentangled representations.** We are hence concerned with *evaluating the usefulness* of disentangled representations (for abstract visual reasoning), rather than *learning* disentangled representations. This distinction is critical: We do not make any assumptions about the *feasibility* of (and *methods* for) learning disentangled representations in the absence of ground-truth factors, and our results have implications for the supervised, semi-supervised, and unsupervised settings. In particular, our results highlight the benefits that current and future research on disentanglement may provide for solving non-trivial upstream tasks that require abstract reasoning.

- **Originality and contribution: novel experimental setup with non-trivial insight.** The research question and the experimental setup is novel and lead to novel insights. Notably on two relevant and non-trivial abstract visual reasoning tasks we find that disentangled representations enable quicker learning using fewer samples. Compare this to a most recent critical work [50], where it could not be observed that higher disentanglement scores reliably lead to a higher sample efficiency on a simple upstream single-factor classification task.

**Additional comments**

We will release all data sets, pre-trained models and code upon publication. Following R3s suggestions we will further improve upon the presentations of results, captions and visualizations.

[Meta-Review · NeurIPS 2019]

This paper aims to evaluate the utility of disentangled representations fo abstract visual reasoning tasks. This is important w.r.t. motivating continued work on learning such representations; the `downstream' utility of these has thus far mostly been taken for granted. This effort aims to put the assumption on steadier footing, via a large-scale empirical analysis over two new tasks that the work introduces. The authors establish that disentanglement empirically leads to sample efficiency, as one would hope. This is an important result to establish because it motivates continued work on learning disentangled representations.